

# Aspirin inhibits tumor progression and enhances cisplatin sensitivity in epithelial ovarian cancer

Jianfeng Guo[*]  Yapei Zhu[*]  Lili Yu  Yuan Li  Jing Guo  Jing Cai  Lin Liu  Zehua Wang

Department of Obstetrics and Gynecology, Union Hospital, Tongji Medical College, Huazhong University of Science and Technology, Wuhan, China
[*] These authors contributed equally to this work.

## ABSTRACT

**Background**. Ovarian cancer is the most common gynecological malignancy and is difficult to manage due to the emergence of resistance to various chemotherapeutic drugs. New efforts are urgently awaited. Aspirin, which is traditionally considered a nonsteroidal anti-inflammatory drug (NSAID), has been reported to exert potential chemopreventive effects. Therefore, we aimed to investigate the anticancer effect and explore the underlying molecular mechanisms of aspirin on epithelial ovarian cancer (EOC) cells.

**Methods**. We conducted wound healing, transwell migration, EdU cell proliferation, colony formation and apoptosis detection assays to observe the effects of aspirin on the migration, proliferation and apoptosis of EOC cells (A2870, Caov-3, and SK-OV-3). EOC cells were treated with a combination of aspirin and cisplatin (CDDP) to observe the effect of aspirin on enhancing CDDP sensitivity. Orthotopic xenograft models of ovarian cancer established with A2780-Luciferase-GFP cells were applied to compare tumor growth inhibition in the control, CDDP and CDDP plus aspirin groups through *in vivo* imaging, which can be used to continuously monitor tumor growth. The expression and acetylation levels of p53 in EOC cells treated with aspirin were determined using western blotting, and p53 acetylation levels were examined in tumors harvested from the transplanted mice. Quantitative real-time PCR was used to assess the mRNA expression of p53 target genes.

**Results**. Aspirin inhibited migration and proliferation and induced apoptosis in EOC cell lines in a concentration-dependent manner. In vitro, aspirin enhanced the sensitivity of EOC cells to CDDP by increasing its inhibitory effect on proliferation and its effect on inducing apoptosis. In vivo, the differences in the tumor growth inhibition rates among the different CDDP experimental groups were statistically significant ($p < 0.05$). Aspirin did not affect p53 protein expression but increased the p53 acetylation level in a concentration-dependent manner. In addition, the mRNA levels of *CDKN1A, BAX, FOXF1, PUMA,* and *RRAD* in EOC cells were significantly increased by the aspirin treatment.

**Conclusions**. Aspirin inhibits tumor progression and enhances the CDDP sensitivity of EOC cells. These antitumor effects of aspirin might be mediated by p53 acetylation and subsequent activation of p53 target genes.

Corresponding authors
Lin Liu, liulinjj@hotmail.com
Zehua Wang, zehuawang@hust.edu.cn

## INTRODUCTION

Ovarian cancer is the most common gynecological malignancy, and in 75% of patients, malignant ovarian tumors are diagnosed at an advanced stage due to the absence of symptoms in early stages (*Siegel, Miller & Jemal, 2015*). Platinum-based combination chemotherapy has been a standard first-line treatment for advanced ovarian cancer, but the development of chemotherapeutic resistance in cancer cells is a major clinical barrier to the successful treatment of cancer (*Szakacs et al., 2006*). *TP53* mutation is a main mechanism inhibiting propagation of the DNA damage signal to the apoptotic machinery, thus leading to the development of drug resistance (*Eckstein, 2011*).

Aspirin (acetylsalicylic acid) is a nonsteroidal anti-inflammatory drug (NSAID); NSAIDs are widely prescribed analgesic and anti-inflammatory agents. Recently, emerging evidence showing the benefits of aspirin in cancer prevention has ignited renewed interest in aspirin research. Data from numerous studies have shown that prolonged intake of aspirin reduces the risk of and improves survival in patients with several cancers, including colon, breast, lung, prostate, and endometrial cancers (*Bardia et al., 2011*; *Lim et al., 2012*; *Huang et al., 2014*; *Cao et al., 2016*; *Chen & Holmes, 2017*; *Takiuchi et al., 2018*; *Gan et al., 2019*; *Loomans-Kropp et al., 2019*). Regular aspirin use is related to a reduction in the ovarian cancer risk (*Trabert et al., 2014*; *Trabert et al., 2019*; *Hurwitz et al., 2020*). Furthermore, *Merritt, Rice & Barnard (2018)* recently assessed the relations between aspirin, other NSAIDs, paracetamol, and ovarian cancer-specific survival after an ovarian cancer diagnosis in approximately 1,000 cases. Their results suggested that women who used aspirin or NSAIDs after diagnosis experienced a prolonged ovarian cancer-specific survival compared with never-users (*Merritt, Rice & Barnard, 2018*). Another small cohort study of women with clear cell ovarian cancer ($n = 77$) reported that aspirin use correlated with longer disease-free and overall survival, and it retained independent significance as a positive prognostic factor (*Wield et al., 2018*).

The mechanism underlying the efficacy of aspirin involves the inhibition of cyclooxygenase (COX) enzymes through irreversible acetylation of serine residues (*Eckstein, 2011*; *Cuzick et al., 2009*; *Robles & Harris, 2010*; *Silwal-Pandit, Langerød & Børresen-Dale, 2017*), leading to inhibited synthesis of prostaglandins (PGs), which cause inflammation, swelling, pain and fever (*Alfonso et al., 2014*; *Lucotti, Ridley & Muschel, 2019*). Moreover, aspirin influences multiple COX-independent biological pathways in tumor progression and metastasis. Aspirin was found to inhibit various signaling pathways, such as the Ras/c-Raf, extracellular signal-regulated kinase (ERK)/MAPK, NF-$\kappa$B, and mTOR pathways, in different cancer cells (*Guo et al., 2016*; *Wu et al., 2020*). Additionally, the chemopreventive mechanism of aspirin might involve the increased number of tumor-infiltrating lymphocytes in the tumor microenvironment. Hence, combination therapy with aspirin might overcome immune resistance and enhance the therapeutic efficacy of chemotherapeutics toward metastatic breast cancer (*Liu et al., 2019*).

Although the molecular mechanisms underlying the anticancer effects of aspirin have been extensively investigated, the involvement of numerous cellular components in ovarian cancer is still incompletely understood. Because of the mutation of *TP53* in platinum-resistant epithelial ovarian cancer (EOC) and the acetylation effect of aspirin, we hypothesized that aspirin exerts an anticancer effect on EOC through the acetylation of p53. Therefore, in this study, we investigated whether aspirin increases the efficacy of cisplatin (CDDP) in EOC and initially explored its underlying mechanism.

## MATERIALS & METHODS

### Cell culture

The EOC cell lines A2780, Caov-3 and SK-OV-3 were purchased from the China Center for Type Culture Collection (CCTCC). A2780 and Caov-3 are ovarian adenocarcinoma cell lines derived from malignant tumor tissues of untreated patients with ovarian cancer. SK-OV-3 is an ovarian adenocarcinoma cell line derived from malignant ascites. All cell lines were maintained at 37 °C in DMEM/F12 (Thermo Fisher, Waltham, MA, USA) supplemented with 10% fetal bovine serum (FBS; Gibco, Carlsbad, CA, USA) in a 5% $CO_2$ atmosphere. Aspirin (Sigma, St. Louis, MO, USA) was added for the indicated times after 12-24 h of cell growth. All assays below were performed in triplicate.

### Cell viability assays

Cells were seeded at a density of 5000 cells/well in 96-well tissue culture plates. Different concentrations of aspirin (0.2 mM, 0.4 mM, 0.8 mM, 1.6 mM, 3.2 mM, or 6.4 mM) were added to the culture media after cell attachment, and five replicate wells for each concentration were established. Forty-eight hours after treatment, 20 µl of 5 mg/ml 3-(4,5-dimethylthiazol-2-yl)-2,5-diphenyltetrazolium bromide (MTT) were added to the media, and the cells were further incubated for 4 h. Subsequently, the cells were solubilized in 100 µl of solubilization solution for 8 h. The optical density (OD) at 570 nm was measured using a microplate reader (Bio-Rad, Berkeley, CA, USA).

### Cell proliferation assays

Cells were seeded in a 96-well tissue culture plate (Corning Costar, Corning, NY, USA) and adjusted to a density of 5000 cells/well. After treatment with different concentrations of aspirin (0 µM, 100 µM, and 1,000 µM) and/or CDDP (20 µM), cells were fixed and stained with an EdU cell proliferation kit (RiboBio Co., Guangzhou, China). Cells were immediately observed under a fluorescence microscope (Olympus, Tokyo, Japan), and cells in three random fields per well were counted (100×).

### Colony formation assays

Cells were seeded in 6-well tissue culture plates in triplicate, and the density was adjusted to 500 cells/well. Cells were cultured in DMEM/F12 supplemented with 10% FBS and different concentrations of aspirin (0 µM, 25 µM, 100 µM, 500 µM, 1,000 µM, and 25 µM) and/or CDDP (5 µM) at 37 °C in a 5% $CO_2$ incubator (Heraeus, Hanau, Germany) for 14 days. Next, cells in the wells were fixed with methanol and stained with 0.1% crystal violet; the cell colonies formed were then visible to the naked eye.

## Flow cytometry analysis

Cells were plated in triplicate in a six-well plate at a density of $1 \times 10^5$ cells/well and incubated for 24 h. Then, cells were treated with different concentrations of aspirin (0 μM, 100 μM, and 1,000 μM) and/or CDDP (20 μM). After a 48-h incubation at 37 °C, cells were harvested, trypsinized and washed with cold phosphate-buffered saline (PBS). Subsequently, cells were stained with APC Annexin V and propidium iodide (PI) (BioLegend, Inc., San Diego, CA, USA). The stained cells were immediately analyzed using an LSR flow cytometer (Beckman Coulter, Inc., 250 S. Kraemer Boulevard, Brea, CA, USA), and the data were analyzed using Summit 6.2 software (Beckman Coulter, Inc., 250 S. Kraemer Boulevard, Brea, CA, USA).

## Western blot analysis

After treatment with different concentrations of aspirin (0 μM, 100 μM, and 1,000 μM) for 24 h, cells were collected in cold Nonidet P40 (NP40) buffer containing a protease inhibitor cocktail for 30 min and centrifuged at $13000 \times$ g for 10 min; subsequently, the supernatants were collected. The protein concentration was measured with a bicinchoninic acid (BCA) assay. Proteins in the samples were separated by 8% sodium dodecyl sulfate-polyacrylamide gel electrophoresis (SDS-PAGE) and transferred onto polyvinylidene difluoride (PVDF) membranes. Membranes were blocked with 5% nonfat milk in Tris-buffered saline containing Tween 20 (TBST) for 1 h at room temperature and incubated with rabbit anti-p53 (1:1500; Cell Signaling Technology, Boston, MA, USA), rabbit anti-acetylated p53 (K382) (1:500; Abcam, Cambridge, UK), and mouse anti-$\beta$-actin (1:4000; Cell Signaling Technology, Boston, MA, USA) antibodies at 4 °C overnight. Horseradish peroxidase-conjugated anti-rabbit or anti-mouse secondary antibodies (1:5000; Cell Signaling Technology, Boston, MA, USA) were used for detection as appropriate. Immunoreactions were visualized with an enhanced chemiluminescence kit (Bio-Rad, Berkeley, CA, USA), and the gray values of the bands were quantified with Quantity One Software (Bio-Rad, Berkeley, CA, USA) or a Molecular Imager ChemiDoc™ XRS $_+$ system with Image Lab™ software (Bio-Rad, Berkeley, CA, USA).

## Quantitative real-time polymerase chain reaction (qPCR)

After treatment with different concentrations of aspirin (0 μM, 100 μM, and 1,000 μM) for 24 h, total cellular RNA was extracted from cultured cells with TRIzol reagent (Thermo Fisher, Waltham, MA, USA) according to the manufacturer's protocol. The quality and quantity of total RNA were evaluated with a NanoDrop 2000/2000C spectrophotometer (Thermo Fisher, Waltham, MA, USA), and cDNAs were subsequently synthesized using a reverse transcription kit (Takara, Tokyo, Japan) and the following primers: *CDKN1A*, upstream 5′-CGATGGAACTTCGACTTTGTCA-3′ and downstream 5′-GCACAAGGGTACAAGACAGTG-3′; *BAX*, upstream 5′-CCCGAGAGGTCTTTTTCCGAG-3′ and downstream 5′-CCCGAGAGGTCTTTTTCCGAG-3′; *FOXF1*, upstream 5′-GCGGCTTCCGAAGGAAAT-3′ and downstream 5′-CAAGTGGCCGTTCATCATGC-3′; *PUMA*, upstream 5′-CAAGTGGCCGTTCATCATGC-3′ and downstream 5′-CAAGTGGCCGTTCATCATGC-3′; *RRAD*, upstream

5′-CAAGTGGCCGTTCATCATGC-3′ and downstream 5′-CAAGTGGCCGTTCATCATGC-3′; and *β-actin*, upstream 5′-GCCAACACAGTGCTGTCTGG-3′ and downstream 5′-GCTCAGGAGGAGCAATGATCTTG-3′. *β*-Actin expression was used for normalization. PCR was performed in an Applied Biosystems StepOnePlus Real-time PCR system (Thermo Fisher, Waltham, MA, USA) using SYBR Green Real-time PCR Master Mix (Takara, Tokyo, Japan). The reactions were performed with a two-step thermal cycling method, 95 °C for 30 s and 40 PCR cycles of 95 °C for 5 s and 66 ° C for 30 s, followed by a melting curve analysis. The relative expression levels were calculated using the Ct ($2^{-\Delta\Delta Ct}$) method.

## Orthotopic xenograft mouse model of human EOC

The Institutional Animal Care and Use Committee of Tongji Medical College approved this research ([2014] IACUC Number: 399). Six-week-old female BALB/c-nu/nu mice were purchased from Beijing HFK Bioscience Co. and were housed in sterile cages with ad libitum access to food in a specific pathogen-free (SPF) barrier animal room at Tongji Medical College Experimental Animal Center. A2780-Luciferase cells were obtained by transfecting the EOC cell line A2780 with a lentivirus carrying the luciferase gene. A2780-Luciferase cells were suspended in serum-free medium, and the density was adjusted to $1 \times 10^7$ cells/ml. A 200-μl aliquot of the cell suspension was subcutaneously injected into the right axilla of nude mice. When the tumor volume reached approximately one cm$^3$, mice were anesthetized with isoflurane, and the tumor was harvested. Subsequently, the tumor was cut into fragments with a volume of approximately 1-2 mm$^3$. Eighteen mice were randomly divided into three groups according to a random number table ($n = 6$). Then, six mice per group were inoculated with one tumor fragment in the left ovary. Mice in the CDDP group were treated with CDDP by intraperitoneal injection and with water by gavage, mice in the CDDP plus aspirin group were treated with CDDP by intraperitoneal injection and aspirin by gavage, and mice in the control group were treated with saline by intraperitoneal injection and water by gavage. Only the administering experimenter was aware of the group allocation. The tumor fragments were transplanted on day 0, and after an intraperitoneal injection of XenoLight D-Luciferin Potassium Salt (Perkin Elmer, Waltham, MA, USA), tumor growth was observed with an IVIS Lumina II (Caliper Life Science, Waltham, MA, USA) beginning on day 8 and then once per week thereafter to obtain *in vivo* bioluminescence images. The acquired optical data were analyzed using Living Image software. Aspirin was initially administered on day 9 at a dose of 20 mg/kg and then every other day thereafter, while CDDP administration was initiated at a dose of 3 mg/kg on day 17 and then every fourth day thereafter. All mice were euthanized via overdose isoflurane inhalation on day 36 and necropsied for the measurement of tumor diameters with Vernier calipers. The tumor volume and relative tumor inhibition rate were calculated using the following formulas:

$$V = (L \times W^2)/2 \text{ and } R = (V_{controlgroup} - V_{treatmentgroup})/V_{controlgroup}.$$

## Immunohistochemistry (IHC)

Tissue sections (4-μm thick) were prepared from formalin-fixed, paraffin-embedded blocks and were immunohistochemically stained with rabbit anti-p53 (1:100; Abcam,

Cambridge, UK), rabbit anti-acetylated p53 (K382) (1:100; Abcam, Cambridge, UK), rabbit anti-Ki67 (1:100; Abcam, Cambridge, UK), rabbit anti-caspase 3 (1:100; Cell Signaling Technology), and rabbit anti-cleaved caspase 3 (1:100; Cell Signaling Technology) antibodies. Localization of the target protein was visualized by incubating sections with a freshly prepared 3,3′-diaminobenzidine (DAB) solution for 3 min. PBS was substituted for the primary antibody as the negative control. Three slides from each tissue were independently evaluated by two observers with an Olympus FV500 optical microscope (Olympus, Tokyo, Japan).

## Quantitative analysis of images of immunohistochemical staining

The expression of target proteins in tumors harvested from transplanted mice was quantified by determining the relative percentage of the positive-stained area to the selected total tissue area and the average staining intensity normalized to the intensity of the selected stroma with an imaging processor to exclude the influence of the tumor volume. Slides were reviewed under a light microscope at 400× magnification, and five regions were randomly selected per slide. ImageJ software was used to analyze the percentage of the positive-stained area and the staining intensity in the tumor tissue from every region, and the overall average value across the slides was considered the level of protein expression.

## Statistical analysis

All statistical analyses were performed using SPSS software version 22.0. Data are presented as the means $\pm$ standard deviations (SD, n $\geq$3) values. Student's $t$-test or one-way ANOVA was adopted to compare group differences, and significance was defined as $p < 0.05$.

# RESULTS

## Aspirin inhibits the growth of human EOC cells *in vitro*

A survival curve based on the results of MTT assays showed that the half-maximal inhibitory concentration (IC50) values of aspirin in the EOC cell lines A2780, Caov-3 and SK-OV-3 were 1.27 (95% CI [0.93–1.72]) mM, 2.05 (95% CI [1.83–2.30]) mM, and 1.54 (95% CI [1.30–21.82]) mM, respectively (Fig. 1A). Thus, we conducted the functional trial *in vitro* with doses of 100 and 1,000 $\mu$M, which were lower than the IC50 in each cell line.

The results of EdU cell proliferation and colony formation assays illustrated that the proliferation and growth of the A2780, Caov-3 and SK-OV-3 cell lines were significantly suppressed by aspirin in a concentration-dependent manner (Figs. 1B–1E and Figs. S2A–S2D). Moreover, as the aspirin concentration increased, the apoptosis rate in the EOC cell lines increased significantly, and the apoptosis rate differed significantly among the different concentrations (Figs. 1F–1G). In addition, the results of wound healing and transwell migration assays showed that the migration of A2780, Caov-3 and SK-OV-3 cells was suppressed by aspirin in a concentration-dependent manner (Figs. S2A–S2D).

## Aspirin enhances the CDDP sensitivity of EOC cells

A2780 and SK-OV-3 cells were treated with CDDP alone or in combination with aspirin to investigate the role of aspirin in mediating CDDP sensitivity *in vitro*. Cell proliferation

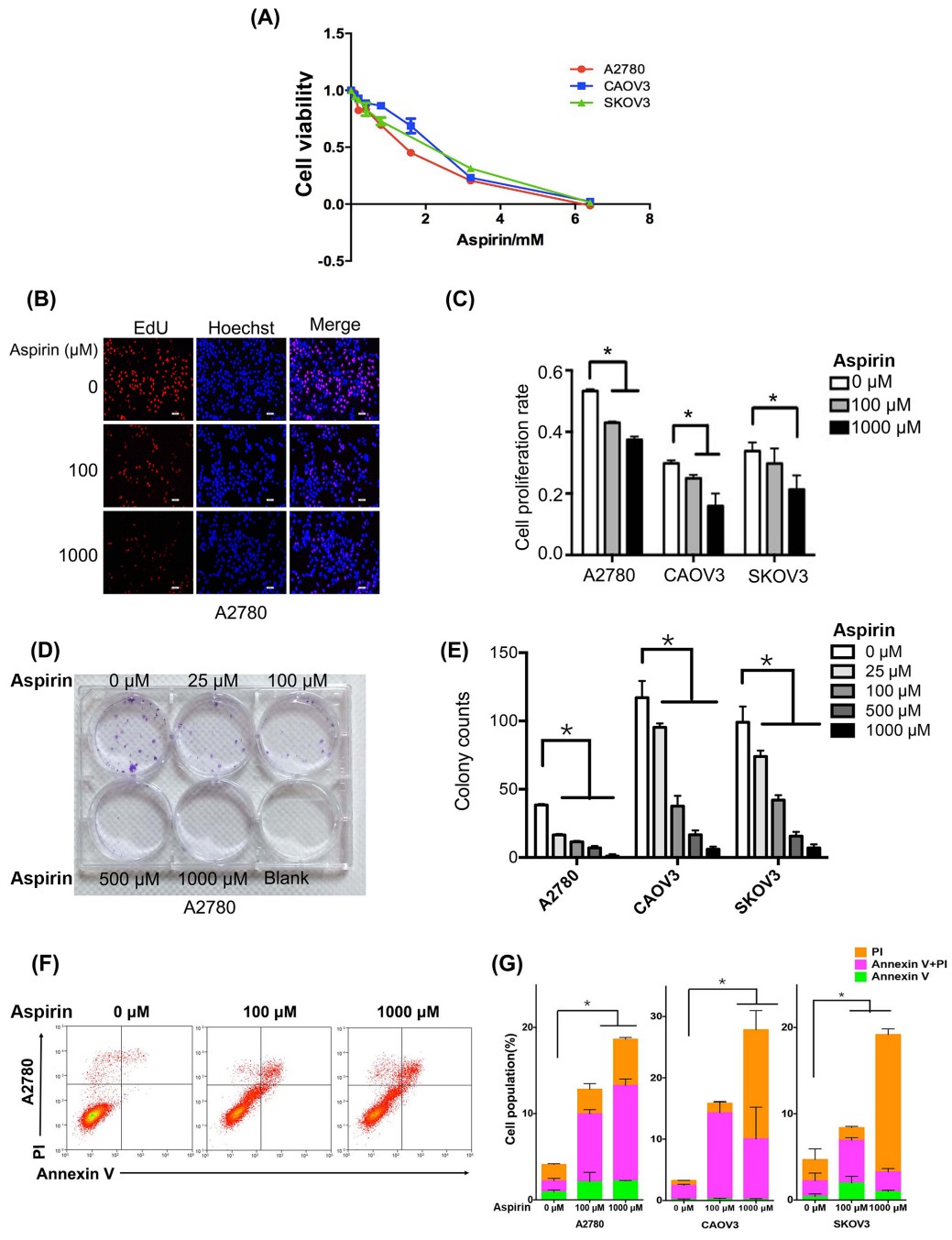

**Figure 1  Aspirin inhibits proliferation and growth and induces apoptosis in human EOC cell lines**
*in vitro.* (A) MTT cell viability assays were performed to determine the IC50s of aspirin in A2780, Caov-3 and SK-OV-3 cells over a 48-h culture period. (B) Cell proliferation was detected using EdU cell proliferation assays (bar, 50 μm) after treatment with different concentrations of aspirin (0 μM, 100 μM, and 1,000 μM) in A2780 cells. (C) Quantitative assay of proliferation rate in A2780, Caov-3 and SK-OV-3 cells. (D) Colony formation assays were conducted after cells were cultured with different concentrations of aspirin (0 μM, 25 μM, 100 μM, 500 μM, 1,000 μM, and 25 μM) for 14 days in A2780 cells. (E) Quantitative assay of colony counts in A2780, Caov-3 and SK-OV-3 cells. (F) Cells were treated with different concentrations of aspirin (0 μM, 100 μM, and 1,000 μM) for 48 h in A2780 cells, stained with the Annexin V-FITC/PI staining kit, and analyzed using flow cytometry. (G) Quantitative assay of apoptosis rate in A2780, Caov-3 and SK-OV-3 cells. Data are presented as the mean ± SD values. * $p < 0.05$.

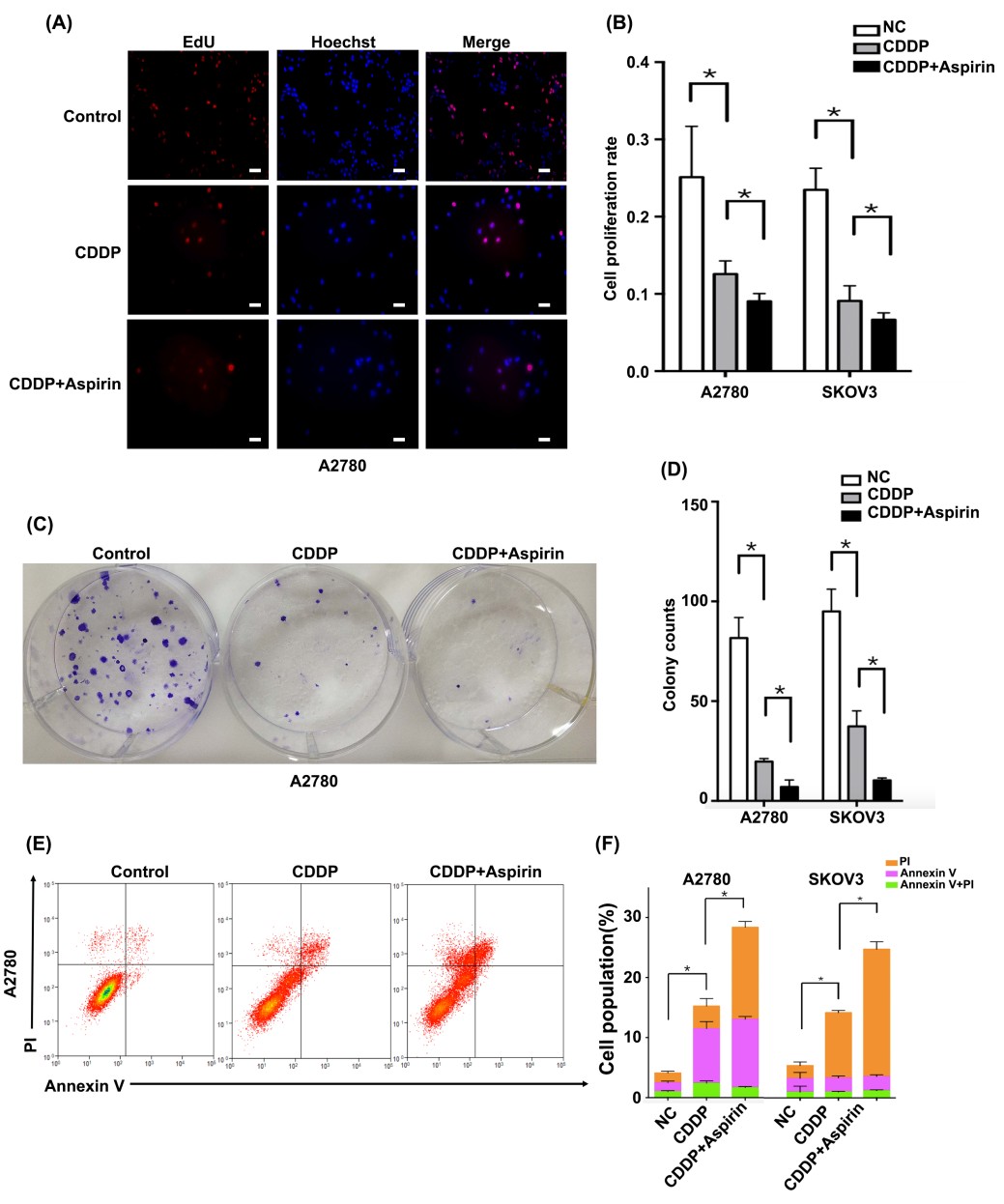

**Figure 2** **Aspirin enhances the CDDP sensitivity of EOC cells *in vitro*.** (A) The cell proliferation rate was determined using EdU cell proliferation assays after the incubation of A2780 cells with CDDP (20 μM) alone or in combination with aspirin (100 μM) (CDDP + aspirin) for 48 h (bar, 50 μm). (B) Quantitative assay of proliferation rate in A2780 and SK-OV-3 cells. (C) Colony formation assays were conducted to observe the growth of A2780 cells after an incubation with CDDP (5 μM) alone or with aspirin (25 μM) for 48 h. (D) Quantitative assay of colony counts in A2780 and SK-OV-3 cells. (E) The percentage of apoptotic cells was determined using flow cytometry after the treatment of A2780 cells with CDDP (20 μM) alone or with aspirin (100 μM). (F) Quantitative assay of apoptosis rate in A2780 and SK-OV-3 cells. Data are presented as the mean ± SD values. * $p < 0.05$.

(Figs. 2A–2B and Fig. S3A) and colony formation (Figs. 2C–2D and Fig. S2B) were significantly inhibited and cell apoptosis (Figs. 2E–2F and Fig. S2C) was induced by CDDP

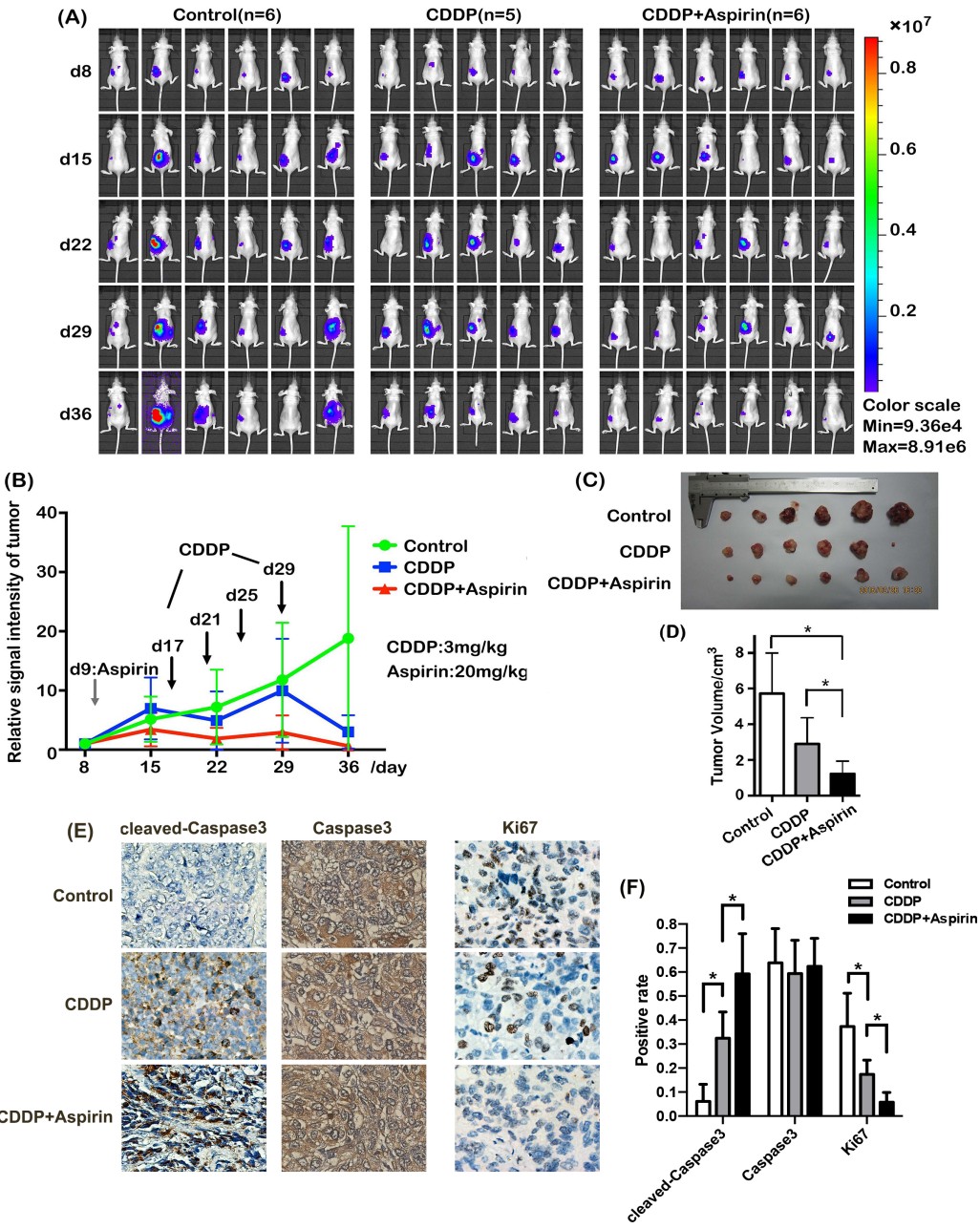

**Figure 3** **Aspirin enhances the CDDP sensitivity of EOC cells *in vivo*.** Orthotopic xenograft mouse models of EOC were established with A2780-Luciferase cells using surgical orthotopic implantation to observe the effects of aspirin *in vivo*. Aspirin was initially administered on day 9 at a dose of 20 mg/kg by gavage and then every other day thereafter, while CDDP administration was initiated at a dose of 3 mg/kg by intraperitoneal injection on day 17 and then every fourth day thereafter. Tumor growth was observed with an IVIS Lumina II beginning on day 8 and then once per week thereafter to obtain *in vivo* bioluminescence images. (A) The tumor signals in all mice (control group, CDDP group, and CDDP + aspirin group) at every time point are displayed. (B) The tumor growth curve for each group (continued on next page...)

**Figure 3 (…continued)**
based on the tumor signals is shown. (C) All mice were euthanized via overdose isoflurane inhalation on day 36 and necropsied for the measurement of tumor diameters with Vernier calipers. (D) Quantitative assay of tumor volume in mice. (E) The expression levels of cleaved caspase 3, caspase 3 and Ki67 in tumor tissues harvested from mice bearing orthotopic EOC xenografts (control group, CDDP group, and CDDP + aspirin group) were assessed using IHC. (F) Quantitative assay for the IHC data of cleaved caspase 3, caspase 3 and Ki67. Data are presented as the mean $\pm$ SD values. * $p < 0.05$.

alone in A2780 and SK-OV-3 cells compared to the corresponding control cells. Moreover, CDDP plus aspirin exerted a significantly stronger effect than CDDP alone.

Orthotopic xenograft mouse models of EOC were established with A2780-Luciferase cells, and tumor bioluminescence signals were visualized *in situ* to investigate the role of aspirin in mediating CDDP sensitivity *in vivo* (Fig. 3A). As shown in Fig. 3B, the tumor growth curve of the CDDP plus aspirin group was shallower than that of the CDDP alone group, and the CDDP alone group had a shallower curve than the control group. Moreover, the difference in the tumor volume among the three groups was consistent with the tumor growth curves (Figs. 3C–3D). The tumor inhibition rates in the CDDP group and the CDDP plus aspirin group were 49.3% and 78.4%, respectively.

In addition, the level of caspase 3 in the tumor tissue was not different among the three groups (control, CDDP, and CDDP plus aspirin); however, the level of cleaved caspase 3 was higher in the CDDP group than in the control group and was higher in the CDDP plus aspirin group than in the CDDP group (Fig. 3E). Ki67 expression was reduced in the CDDP group compared to the control group and was lower in the CDDP plus aspirin group than in the CDDP group (Fig. 3E).

## Aspirin acetylates p53 in EOC cells and subsequently activates p53-mediated downregulation of target genes

The expression of p53 was not different among the three groups (control, CDDP, and CDDP plus aspirin), but the acetylation level of p53 in the CDDP plus aspirin group was higher than that in both the control group and the CDDP group (Figs. 4A–4B). As shown in Fig. 4C, the level of the p53 protein in the A2780 cell line remained unchanged after treatment with different concentrations of aspirin. However, as the aspirin concentration increased, the acetylation level of p53 gradually increased. Moreover, the mRNA expression levels of the p53 target genes CDKN1A, BAX, FOXF1, PUMA, and RRAD increased significantly in a concentration-dependent manner, as illustrated in Fig. 4D.

## DISCUSSION

The anticancer action of aspirin has recently received increasing attention. An inverse association between the intake of aspirin and the risk of many types of cancer, including ovarian cancer, has been identified in recent decades (*Takiuchi et al., 2018*; *Gan et al., 2019*; *Loomans-Kropp et al., 2019*; *Baandrup et al., 2015*; *Peres et al., 2016*; *Zhang et al., 2016*). In addition to its use as an effective chemopreventive agent, aspirin can also prolong patient survival even after a cancer diagnosis, which suggests the potential for the adjunct use of aspirin in cancer chemotherapy (*Merritt, Rice & Barnard, 2018*). However, despite these

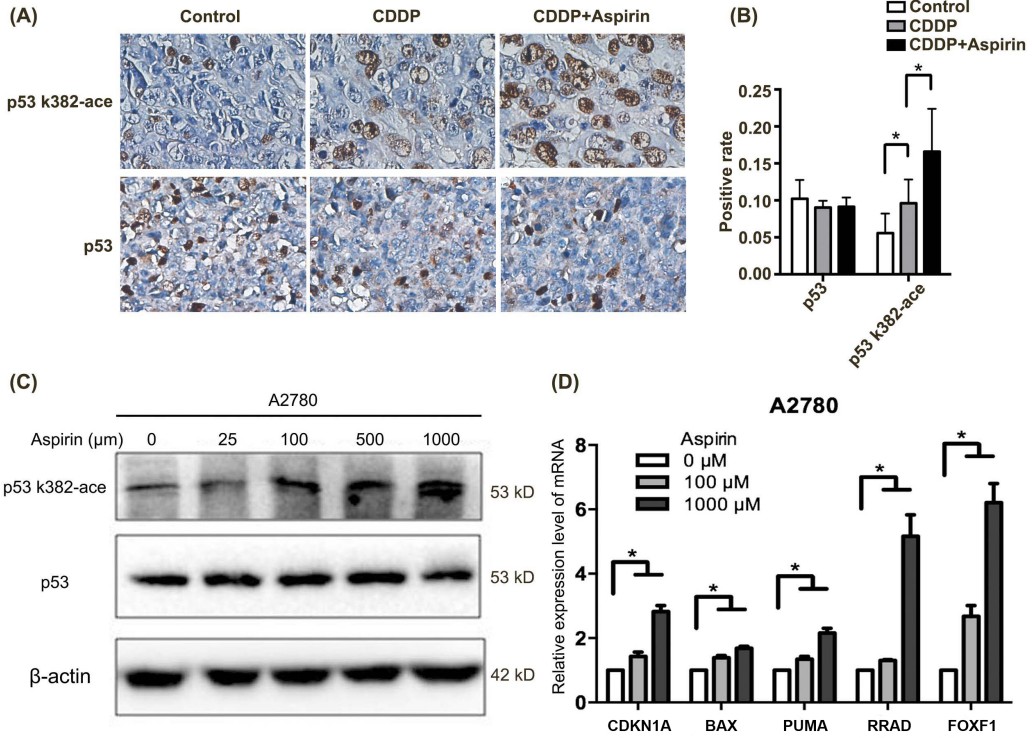

**Figure 4** **Aspirin acetylates p53 in A2780 cells and subsequently activates p53-mediated downregulation of target genes.** (A) The expression of p53 and acetylation level of p53 in tumor tissues from mice bearing orthotopic EOC xenografts (control group, CDDP group, and CDDP + aspirin group) were assessed using IHC. (B) Quantitative assay of positive rate in IHC. (C) Western blot analysis was conducted to evaluate the level of the p53 protein and its acetylation (p53 K382-ace) after treatment with different concentrations of aspirin (0 µM, 25 µM, 100 µM, 500 µM and 1,000 µM). (D) Levels of the p21, bax, foxf1, puma, and rad mRNAs were examined using qPCR after treatment with different concentrations of aspirin (0 µM, 100 µM, and 1,000 µM). Data are presented as the mean ± SD values. * $p < 0.05$.

observations and reports, the mechanisms underlying the anticancer effects of aspirin are not well understood. In the present study, we demonstrated that both *in vitro* and *in vivo*, aspirin could inhibit tumor cell growth and mobility and enhance their sensitivity to CDDP, which is similar to the previous findings that the combination therapy with aspirin overcome immune resistance and enhance the therapeutic efficacy of chemotherapeutic agent toward metastatic breast cancer. (*Liu et al., 2019*).

Since plasma concentrations of intact aspirin can be up to 100 µM after the administration of a single therapeutic dose (*Roberts et al., 1983*), this concentration was used to observe the cellular effects of aspirin in the present study. The results of wound healing, transwell migration, EdU cell proliferation and colony formation assays showed that aspirin inhibited the migration, proliferation and growth of A2780, Caov-3 and SK-OV-3 EOC cells, consistent with the results of previous studies (*Huang et al., 2016*).

Animal studies have indicated that aspirin significantly enhances the efficacy of CDDP in reducing the growth of orthotopic xenograft tumors formed by human EOC cells in mice. Notably, because of the gastrointestinal side effects of this drug, as well as the potential for

gastroduodenal ulcers and bleeding, we treated mice with a safe dose of 20 mg/kg, which was determined by an equivalent dose conversion from the dosage of 100 mg/day used for oral low-dose aspirin therapy in humans. Moreover, in the present animal study, we established orthotopic xenograft models in mice (*Kim et al., 2006*) using a luciferase-labeled A2780 cell line and subsequently observed tumor growth using *in vivo* bioluminescence imaging after an intraperitoneal injection of D-luciferin potassium salt, which interacts with luciferase to generate an optical signal. Unlike previous studies using models established by an intraperitoneal injection of ovarian cancer cells or using subcutaneous tumor models (*Huang et al., 2016*), we surgically transplanted the tumor tissue into the ovaries of a mice; thus, our model more precisely mimicked the environment of human ovarian cancer. In addition, this approach enabled continual observation of tumor growth deep inside the abdominal cavity via *in vivo* imaging without the need to sacrifice the mice. Moreover, due to the principle of *in vivo* bioluminescence imaging, which detects the viable cells in the tumor, our approach provided a more accurate estimate of tumor activity than simply measuring the tumor volume. In summary, the results of our animal experiments were more convincing and reliable than those reported in other studies.

Regarding the potential mechanism, we elucidated that aspirin acetylates p53 at lysine 382 in EOC cells and subsequently activates p53 target genes, which regulate cell cycle arrest, migration and apoptosis.

Aspirin consists of two functional groups, an acetyl group and a salicylate group, both of which have their own distinct targets. The acetyl group of aspirin acetylates COX, thereby inducing its inhibition (*Alfonso et al., 2014*), and the salicylate group suppresses I$\kappa$B kinase (IKK)$\beta$ and prevents NF-$\kappa$B activation (*Kim et al., 2006*); thus, both groups contribute to the anti-inflammatory properties of aspirin. Although most hypotheses have focused on the anti-inflammatory activities of aspirin, this drug also interferes with ERK signaling, leading to ERK inhibition by preventing the binding of c-Raf to Ras *in vitro* (*Guo et al., 2016*). Moreover, aspirin reduces mTOR signaling by inhibiting the mTOR effectors S6K1 and 4EBP1 in colorectal cancer (*Guo et al., 2016*) and suppresses esophageal squamous cell carcinoma growth by inhibiting HBXIP and HMGA2 (*Wu et al., 2020*). In addition, aspirin induces concentration-dependent inhibition of the major oncogenic pathway, the Wnt/$\beta$-catenin pathway, in colon cancer (*Bos et al., 2006*) and acetylates the tumor suppressor protein p53 at lysine 382 in MDA-MB-231 breast cancer cells (*Alfonso et al., 2009*).

*TP53* has been recognized as a tumor suppressor gene encoding the p53 protein in human cancers. The p53 protein is a transcription factor that controls the expression of genes and miRNAs affecting many important cellular processes. The chemotherapy drug CDDP activates the DNA damage response pathway, in which the tumor suppressor p53 controls cell cycle progression and apoptosis. However, studies have reported that the *TP53* gene is mutated and inactivated in more than 50% of all tumors and that these mutations are associated with earlier onset of cancer and the development of drug resistance (*Robles & Harris, 2010*). A considerably high mutation frequency of 50–100% is reported across all ovarian cancers (*Silwal-Pandit, Langerød & Børresen-Dale, 2017*). Meanwhile, p53 mutation can prevent the proper folding of the p53 protein or directly disrupt its DNA-binding ability, and thus the tumor may acquire new oncogenic properties such

as drug-resistance (*Raimundo et al., 2020*). Posttranslational modifications (PTMs) of p53 are critical in modulating its tumor suppressor functions (*Vousden & Prives, 2009*). Acetylation is an important PTM that increases p53 protein stability, its binding to low-affinity promoters and its interactions with other proteins (*Reed & Quelle, 2014*). For instance, p53 acetylation at K382 promotes the recruitment of its co-activators to the *CDKN1A* promoter and increases histone acetylation following DNA damage (*Reed & Quelle, 2014*); it also activates pro-apoptotic functions of BAX and PUMA (*Yamaguchi et al., 2009*). Moreover, RRAD and FOXF1 are also upregulated by p53 activation, which may further inhibit cell migration (*Hsiao et al., 2011*; *Tamura et al., 2013*). In our study, aspirin acetylated p53 to subsequently increase the mRNA expression of its target gene *CDKN1A* in A2780 and SK-OV-3 EOC cells. Furthermore, aspirin activated other downstream target genes related to cell migration, such as *RRAD* and *FOXF1*, and genes related to apoptosis, i.e., *PUMA* and *BAX*. However, more studies are needed to confirm whether the anticancer efficacy of aspirin is mediated by this pathway.

According to previous research, A2780 and SK-OV-3 cells express the wild-type *TP53* gene, while the *TP53* gene is mutated in Caov-3 cells, which is a point mutation that resulted in a chain termination signal likely to truncate the p53 peptide at amino acid 135 (*Skilling et al., 1996*; *Yaginuma & Westphal, 1992*). The status of the *TP53* gene in these three EOC cell lines was not validated in our study. Moreover, further research is required to investigate the potential differences in biological function between the acetylation of wild-type p53 versus mutant p53 in ovarian cancer. Beside p53 acetylation, as a multifunctional agent, Aspirin may exhibit anticancer effects through p53-independent pathways, such as inhibition of cyclin dependent kinases by aspirin metabolites and inhibition of nuclear-$\kappa$B signaling through direct interactions with I$\kappa$B kinase in colorectal cancers (*Sankaranarayanan et al., 2020*). The multi-targeting and dose-dependent action manner of aspirin makes its use in the treatment of cancer both promising and puzzling. Although we provide preliminary evidence regarding the potential benefit of adding aspirin to CDDP chemotherapy in ovarian cancer, the interactions between aspirin and traditional chemotherapeutic agents remains largely unclear.

In summary, aspirin inhibits tumor progression and enhances the CDDP sensitivity of EOC cells through p53 acetylation and subsequent activation of p53 target genes that regulate tumor migration, proliferation, and chemoresistance. These findings provide a theoretical basis for the use of aspirin in the treatment of ovarian cancer and lay the foundation for future research.

### Funding

This work was supported by the National Natural Science Foundation of China (No. 81672573). The funders had no role in study design, data collection and analysis, decision to publish, or preparation of the manuscript.

## Grant Disclosures

The following grant information was disclosed by the authors:
National Natural Science Foundation of China: 81672573.

## Competing Interests

The authors declare there are no competing interests.

## Author Contributions

- Jianfeng Guo conceived and designed the experiments, analyzed the data, prepared figures and/or tables, and approved the final draft.
- Yapei Zhu and Jing Guo performed the experiments, prepared figures and/or tables, and approved the final draft.
- Lili Yu analyzed the data, authored or reviewed drafts of the paper, and approved the final draft.
- Yuan Li performed the experiments, authored or reviewed drafts of the paper, and approved the final draft.
- Jing Cai conceived and designed the experiments, prepared figures and/or tables, authored or reviewed drafts of the paper, and approved the final draft.
- Lin Liu and Zehua Wang conceived and designed the experiments, authored or reviewed drafts of the paper, and approved the final draft.

## Animal Ethics

The following information was supplied relating to ethical approvals (i.e., approving body and any reference numbers):

The Institutional Animal Care and Use Committee of Tongji Medical College approved this research ([2014] IACUC Number: 399).

## Data Availability

The raw measurements are available in the Supplementary File.

## Supplemental Information

Supplemental information for this article can be found online at http://dx.doi.org/10.7717/peerj.11591#supplemental-information.

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
