# Peer review of "Aspirin inhibits tumor progression and enhances cisplatin sensitivity in epithelial ovarian cancer"

_PeerJ, doi:10.7717/peerj.11591_

## Round 0.1 · original submission · Minor Revisions

Your manuscript,"Aspirin inhibits tumor progression and enhances cisplatin sensitivity in epithelial ovarian cancer," has been reviewed by two expert reviewers and an academic editor. Our comments are presented below. The reviewers indicate that the topic could be of interest to the field, but note several issues that need attention before further consideration. One reviewer recommended rejection, so I urge your careful consideration of their comments as you revise your manuscript.

Both reviewers noted significant omissions in the cited literature and a lack of consideration for mechanisms beyond p53 acetylation for the observed phenotypes, which impacted their basic reporting score. Reviewers also noted the presence of grammatical errors throughout the manuscript and requested more clarity in figure legends to improve readability. PeerJ does not provide copy editing services, so it is critical that the accepted manuscript be well written.

I look forward to receiving your revised work.

Reviewer 1 ·

Basic reporting

Throughout the text, there are grammatical errors. Please make sure that the sentences are scientifically sound and grammatically correct.
More updated literature can be cited.

Experimental design

The experimental design is appropriate. Methods described are sufficient. Rigorous investigations performed.

Validity of the findings

The results presented are interesting and conclusions well written.

Additional comments

It is reported that ovarian cancer is a common gynecological malignancy, and it is difficult to treat it because these cancers often exhibit chemoresistance. The authors in this paper studied the effect of aspirin alone or in combination with cisplatin (CDDP) on epithelial ovarian cancer cells (EOC) cell proliferation, cell migration, colony formation, apoptosis etc. The investigated cell lines include: A2870, Caov-3 and SK-OV3. The authors demonstrated that aspirin treatment inhibited the migration, proliferation in these cells and also induced apoptosis. Interestingly, aspirin potentiated the effect of CDDP. The authors demonstrated that aspirin induces acetylation of p53; however, does not affect the expression levels of p53. This was associated with increased expression of p21, Bax, Foxf1, PUMA and Rad proteins in these cell lines. They conclude that aspirin’s action on tumor regression and enhancement of CDDP may occur through acetylation of p53 and induction of p21, Bax, etc.

Comments:
1. The authors state that the A2780 and SK-OV-3 cells contain wild type p53 and Caov-3 cells contains mutant p53. It appears that from the results presented, aspirin acetylates both wild type and mutant p53. What is the p53 mutation (amino acid replacement) reported in Caov-3 cells?
2. P53 contains several lysine residues that are potentially acetylated by aspirin. The antibodies used in the study detected K382. It will be useful to discuss the significance of K382 acetylation for gene expression if that information is available in literature.
3. The authors emphasized on the potential role of p53 acetylation in aspirin’s anti-cancer effects. They should open to the possibility that other pathways may also contribute to aspirin’s anticancer effects. In this regard, citing a recent review article published by Sankaranarayanan et al (International Journal of Molecular Sciences, Int. J. Mol. Sci. 2020, 21(23), 9018; https://doi.org/10.3390/ijms21239018 is appropriate.
4. Throughout the text, there are grammatical errors. Please make sure that the sentences are scientifically sound and grammatically correct.

Reviewer 2 ·

Basic reporting

.

Experimental design

.

Validity of the findings

.

Additional comments

The manuscript entitled “Aspirin inhibits tumor progression and enhances cisplatin sensitivity in epithelial ovarian cancer” investigated the effects of aspirin on several ovarian cell lines and cisplatin-treated cells. Besides, its tumor effect on tumor-bearing rodent was evaluated. Finally, the acetylation status of p53 was examined. In conclusion, aspirin can inhibit tumor progression and enhance the CDDP sensitivity of EOC cells. These antitumor effects of aspirin might be mediated by p53 acetylation and subsequently activation of p53 target genes.

Besides its effect on COXs, antitumor effect of aspirin is well reported in many malignancies, including ovarian cancer. Not only aspirin but also aspirin derivatives are demonstrated to have antitumor effects and sensitize tumor cells to chemotherapeutic treatments including cisplatin. Thus, current study focuses and findings are not new to the readers.

1. There are studies demonstrating the effects of NSAIDs, aspirin, and aspirin derivatives on malignancies, including ovarian cancer. Unfortunately, those relevant studies and findings were not described and mentioned in this manuscript.
2. Figure legends should be checked for the clear description of experimental protocols and conditions.
3. Figure 2 and figure 3, a group of aspirin alone should be included for comparison.
4. Current study only measured the change of p53 acetylation after treatment in the presence of aspirin, well-known phenomena. The specific involvement and action of p53 should be determined.
5. Overall, current findings were not enough to provide additional insights into the study of aspirin and ovarian cancer. Advanced studies are needed.

---

## Round 0.2 · Minor Revisions

Please include discussion of what if anything is known regarding p53 acetylation at K382 for rad and foxf1 expression in response to Reviewer 1 comments.

Quantitation should be provided for the IHC data presented in Fig 3E.

Micrographs in Fig 2A are very dark and difficult to appreciate as presented. Please revise.

Add Aspirin to the figure panels of Figure 1C-G.

Please review manuscript for proper HUGO nomenclature for gene names (see lines 269-70, 554-555, Figure 4). Note this is not a comprehensive list, just several noted during review.

---

## Round 0.3 · accepted · Accept

I have reviewed the rebuttal letter and find that the authors have addressed critiques sufficiently. The manuscript is acceptable for publication.